# Impact of Land Property Rights Security Cognition on Farmland Quality Protection: Evidence from Chinese Farmers

Laiyou Zhou [1,2], Hua Lu [1,*] and Jinlang Zou [1]

1  Institute of Ecological Civilization, Jiangxi University of Finance and Economics, Nanchang 330013, China
2  School of Economics and Management, Xinyu University, Xinyu 338004, China
*  Correspondence: luhua@jxufe.edu.cn

**Abstract:** The stability or security of property rights plays an important role in stimulating the investment of economic entities, which can prevent or alleviate the degradation of land resources and improve the efficiency of agricultural management. This paper focuses on the perspective of land property rights security awareness, analyzes its impact on farmers' cultivated land quality protection behavior, and, on the basis of a theoretical analysis framework linking the two, uses farmer survey data from southern Jiangxi Province and the probit model to empirically analyze the impact of farmers' land property rights security awareness on their cultivated land quality protection behavior. The results show that the improvement in farmers' awareness of land property rights security could significantly improve their probability of applying farmyard manure. This conclusion is proven to be robust by means of a replacement model and adjustment variables. Moreover, it is found that the influence of land property rights security cognition on farmyard manure application differs among farmers by age, degree of part-time work and land scale. Finally, we can draw inspiration for improving the effective implementation of public policies and increasing the publicity of legal knowledge of land property rights to improve farmers' awareness and consideration of land property rights security.

**Keywords:** property rights security; cognition; cultivated land quality; protection; investment

## 1. Introduction

High-quality cultivated land is the basis for solving the problem of agricultural land degradation and achieving sustainable agricultural development [1]. In the context of the continuous growth of the population and the continuous reduction in the amount of cultivated land, the protection of the quality of cultivated land has attracted increasing attention from government departments and academia. According to Guangming.com, due to intensive utilization, the basic fertility of cultivated land has declined for a long time. In 2018, the amount of chemical fertilizer used was 6.4 times that in 1978, while the grain yield was only 2.2 times. Therefore, the protection of cultivated land quality is important. In China, the stakeholders of cultivated land quality protection mainly include the central government, local governments at all levels, village collectives and farmers [2]. To encourage all kinds of stakeholders to actively participate in the protection of cultivated land quality, the Chinese government has issued a series of policies and documents to encourage and constrain the use of cultivated land by all stakeholders. For example, as early as 2015, the Ministry of Agriculture issued the Action Plan for the Protection and Improvement in Cultivated Land Quality, which proposed major actions such as comprehensive treatment of degraded cultivated land, remediation of contaminated cultivated land, and improvement in soil fertility protection. The purpose of these actions is to improve the internal quality of cultivated land, realize "storing food on the ground", and consolidate the foundation of national food security [3]. As the most direct user of cultivated land, farmers' cultivated land use behavior has an important impact on the

quality of cultivated land. Therefore, how to encourage farmers to protect the quality of cultivated land is a topic of great scholarly interest.

Existing research has found that the factors affecting farmers' cultivated land quality protection behavior are multifaceted and are found at the institutional level, the family level, the farmer level, and the plot level [4–7]. Among the institutional factors, the security or stability of agricultural land property rights is an important factor that has been widely mentioned. Research shows that it has a significant impact on farmers' cultivated land quality protection behavior. Stable property rights can form the expectation of "definite investment income" for farmers and then encourage them to invest in protecting cultivated land; furthermore, stable property rights are more likely to encourage farmers to use land sustainably without excessive or even predatory use of land, which to some extent promotes their investment in farmland protection [1].

Although the research on the impact of the security or stability of agricultural land property rights on the protection of cultivated land quality is rich, most studies focus on the impact of objective legal systems on farmers' cultivated land quality protection behavior [8,9]. In fact, legal and practical land property security impacts farmers' behavior mainly through their awareness of land property security. Of course, subjective property rights cognition does not negate the role of the objective property rights institutional arrangement. However, due to the differences in farmers' cognitive level, knowledge, experience, preferences and other factors, their subjective property rights cognition may differ when facing the same property rights institutional arrangement. According to a survey on the perception of rural land rights in 17 provinces of China, 39.3% of farmers believe that rural land is owned by the state or the government, 32.7% believe that it is owned by farmers, 8.5% do not know the ownership of agricultural land, and 19.5% say that agricultural land is collectively owned by villages (groups) [1].

In addition, research shows that farmers' awareness of land property rights security is an important factor that has a significant impact on their behaviors, such as land transfer decisions and land acquisition disputes [8,10–12]. However, studies considering the impact of farmers' awareness of land property rights security on their cultivated land quality protection behavior are scarce, and few studies have deeply explored the impact mechanism of farmers' awareness of land property rights security on cultivated land quality protection, which provides impetus for this study. Therefore, this paper uses survey data from micro farmers in Jiangxi Province, adopts the probit model to carry out benchmark regression analysis, discusses the endogeneity of the benchmark regression model, conducts robustness tests through the replacement model method, adjustment variables and other methods, and analyzes the heterogeneous impacts on the application of farm manure by farmers with different endowment characteristics.

## 2. Theoretical Analysis and Research Hypothesis

According to property rights theory, stable agricultural land property rights can form a stable investment return expectation, reduce the occurrence of uncertainty, and increase investment activity [13]. From this point of view, stable agricultural land property rights are essential to mobilize the enthusiasm of farmers' production and management and encourage farmers' investment behavior in farmland protection. Studies in China and abroad have confirmed this view. For example, Wannasai et al. [14] analyzed the land use behavior of farmers in the Prasae watershed, Thailand, and found that the insecurity of property rights is directly related to forest occupation and destruction. They believed that safer land property rights would be beneficial to production and long-term investment. Nizalov et al. [15] took the Ukrainian land market as an example to study the relationship between land leasing and land investment and found that inadequate protection of land use rights would lead to insufficient investment in capital-intensive land. In rural Burkina Faso, researchers found that the relationship between land property rights and land investment is mutually reinforcing. That is, stronger land property rights can increase farmers' investment in land, and land investment can further strengthen farmers' land property

rights [16]. Cao et al. [17] found that the security of land property rights has a significant positive impact on the probability of farmers' investment in cultivated land, especially investment in the quality protection of private land. Qian et al. [2,18,19], based on a survey of farmers in the Guangxi Zhuang Autonomous Region of China, found that the new round of land ownership determination can significantly encourage farmers to adopt behaviors such as soil fertility improvement and nutrient balance and effectively encourage them to adopt more behaviors to protect the quality of cultivated land. However, land adjustment experience weakens the incentive effect of land protection investment brought by land ownership determination. Many studies have used data at the household or plot level to find a significant positive relationship between property rights security and investment in farmland protection [20–24].

Even so, some studies have found that there is little or no significant relationship between land property rights security and land protection investment [25,26]. Rao et al.'s [27] research on 325 interplanting farmers in Xinjiang, China, found that even with the official land certificate, farmers could not form a safe understanding of land property rights because the farmers with the official land certificate showed no significant impact in terms of investment in contract land and wasteland. Evidence from Ethiopia shows that insecurity of land property rights is not necessarily related to investment in land protection. Even the insecurity of land property rights does not affect poor farmers' investment in land to prevent land degradation [28]. Some studies from other fields also found that there is no causal relationship between property rights and investment. For example, the Energy Charter Treaty (ECT) aimed at strengthening property rights did not improve the investment environment [29], the implementation of the rural land regularization project in Benin did not improve investment in soil fertility [30], and the security reform of forestland property rights in Nicaragua unexpectedly increased deforestation [31]. The academic community lacks consensus on the relationship between land property security and land protection investment. This is because the regions studied and data used are different [1] and because different indicators are used to measure the stability or security of property rights. Some studies use legal property rights security indicators, such as land certificates, some use objective property rights security, and others use perceived property rights security [32].

There is also much discussion on the impact of land property rights security cognition on farmers' production and operation. Thu et al. [33] used the data of 1834 farmers in Vietnam to verify the relationship between farmers' awareness of land property rights security and organic fertilizer application after the new land law was passed in 2013. The results showed that after the implementation of the new land law, farmers' awareness of land property rights security was enhanced, which had a significant positive impact on organic fertilizer application. Research on farmers in Guangxi, China, found that the new round of land ownership confirmation was conducive to improving farmers' perception of land property rights security, which further encouraged farmers to take diversified measures to protect the quality of cultivated land. Among them, the probability of applying organic fertilizer increased by 18.42%, the probability of returning straw to the field increased by 18.28%, and the probability of subsoiling increased by 18.70% [2]. Using microdata from eight provinces in China, Su [1] analyzed the protective investment behavior of farmers' organic fertilizer application under different degrees of property rights awareness. The results showed that the amount of organic fertilizer applied by farmers with strong property rights awareness was significantly higher than that of farmers with weak property rights awareness.

In fact, land property security influences farmers' behavior at the legal and factual levels through their awareness of land property security [34,35]. Broegaard [36] believed that the perception of property rights formed the basis for farmers' decisions and actions. Only by taking farmers' perception of property rights as the central factor for analysis can farmers' behavior be well understood [1]. Of course, this does not imply that the legal and factual security of property rights is unimportant. In contrast, the cognition of farmers' security of land property rights is derived from the legal or factual security of

property rights. Otherwise, simply discussing the concept and connotation of property rights security cognition would be meaningless.

Furthermore, this paper divides farmers' land property rights cognition into "strong property rights cognition" and "weak property rights cognition". When farmers think that contracted farmland belongs to them, it can be considered strong property rights cognition; when farmers think that the contracted farmland is owned by the village collective or the state, it can be considered weak property rights cognition. The reason for this division is that although the Chinese government confirmed and issued certificates for farmland contracted by farmers in 2013 and the issuing rate exceeded 96% [2], due to differences in implementation, farmers' perception and preference levels, and the existence of different processes of land adjustment, farmers have different perceptions of land property rights.

Farmers with strong property rights awareness believe that their investment income in their farmland is protected by law and will not be taken by others. Thus, they form a stable expectation of future income, which further encourages them to invest in farmland. To a certain extent, this can promote the protective investment of their cultivated land because if the farmers only use the land and do not protect it, or if they even operate predatorily, they, the property owners, will bear the consequences. In contrast, farmers with a weak property rights perception believe that the ownership of contracted land belongs to the village collective or the state. When making production and management decisions, these farmers pay more attention to short-term interests, tend to overuse or develop the land, and rarely invest in the long-term protection of cultivated land, ignoring the sustainable use of cultivated land. See Figure 1 for the above relationship between land property rights awareness and investment in the protection of cultivated land quality:

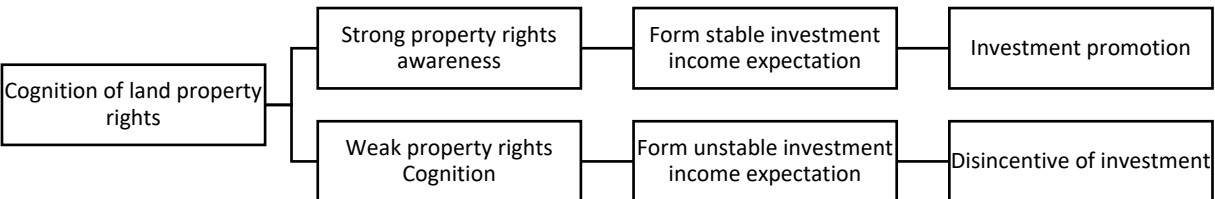

**Figure 1.** Theoretical framework of the impact of land property rights security cognition on farmland protection investment.

In summary, this paper proposes the following hypothesis: farmers' awareness of land property rights affects their cultivated land protective investment behavior. A strong awareness of property rights promotes cultivated land protective investment, and a weak awareness of property rights inhibits cultivated land protective investment.

### 3. Method

*3.1. Model*

Cultivated land quality protection investment refers to productive investment that is conducive to maintaining or improving the quality of cultivated land or barren land, such as returning straw to the field, planting green manure, applying farmyard manure and commercial organic fertilizer, engaging in soil testing and formulated fertilization, seeking to improve the ability of soil to retain water, adding fertilizer, renovating canals, leveling land and engaging in deep tillage [37]. In this paper, the farmers in the study area mainly use farmyard manure to improve the fertility of cultivated land and protect the land quality. Whether farmers apply farm manure is a binary variable, so a binary probit model is selected for estimation. The specific model is as follows:

$$y = a_0 + a_1 cognition + \sum \beta_i x_i + \varepsilon_1$$

In the equation, *y* refers to farmers' cultivated land quality protection behavior; *cognition* indicates farmers' awareness of land property rights security; $x_i$ represents a

series of control variables; $a_0$ is a constant term; $a_1$, $\beta_i$ is the variable coefficient; and $\varepsilon_1$ represents a random perturbation term.

### 3.2. Endogeneity Problems

Theoretically, the benchmark regression model may have endogeneity problems [16]. First, there may be missing variables in the model if there are variables related to farmers' perception of land property rights security among the uncontrolled variables that affect cultivated land quality protection behavior. Second, the model may have two-way causality. That is, farmers' awareness of land property rights security may affect their farmland quality protection behavior, while their investment in farmland quality protection may further affect their awareness of land property rights security. However, some scholars believe that the act of swearing sovereignty over agricultural land investment occurs only on public land without clear property rights, while the problem of endogenous property rights generally does not arise in agricultural land contracted by farmers [38]. To further test whether the benchmark regression model has endogeneity problems, this paper uses the method of Ma et al. [39] for reference and selects the average land property rights security perception of other (n-1) surveyed farmers in the village as the instrumental variable of farmers' land property rights security perception to address any potential endogeneity problem in the model. The basis for selecting this instrumental variable is as follows. First, the communication of information among farmers improves the understanding of the farmers in the village, and the land property rights security awareness of the surveyed farmers in the same village reflects the land property rights security awareness of the village, which is closely related to the land property rights security awareness of individual farmers in the village. Second, excluding farmers' awareness of land property rights security, the selected instrumental variables are not directly related to farmers' investment in farmland quality protection.

## 4. Data and Variable Selection

### 4.1. Data Source

To investigate the impact of farmers' awareness of land property rights security on cultivated land quality protection behavior, the research team took the Gannan Region, Jiangxi Province, as an example (see Figure 2) and conducted a farmer survey from July to September 2021. The terrain of the Gannan Region is complex, mainly including mountains, hills and basins, with large regional differences. The survey found that even after a new round of land ownership confirmation, some towns still have sporadic land adjustment. According to the regional topographic characteristics and the principle of random sampling, the survey selected five representative counties and cities, namely, Suichuan County, Yudu County, Ruijin City, Xinfeng County and Xunwu County. Four towns were randomly selected in each county and city, and 3–5 natural villages were selected in each town. Then, the number of respondents was determined according to the natural village's resource endowment (population, land area, regional characteristics, etc.). After deleting some questionnaires with missing key data, data for 669 farmers were obtained in this survey.

### 4.2. Variable Selection

The variables in this paper are roughly divided into three categories: explained variables, explanatory variables, and control variables. The definition or assignment of each variable is as follows:

#### 4.2.1. Interpreted Variable

As mentioned above, the quality protection behaviors of cultivated land are diverse and include the return of straw to the field, farmyard manure application, deep tillage and land leveling. The study area in this paper has complex terrain, including hills and basins. It is difficult to carry out investment activities such as land leveling and deep cultivation under unified planning. In view of the characteristically poor soil, most farmers

in the study area choose to use farm manure to maintain soil fertility and achieve farmland quality protection. Therefore, this paper takes whether farmers apply farmyard manure as the explained variable; that is, if farmers apply farmyard manure, the value is yes = 1, and if farmers do not apply farmyard manure, the value is no = 0.

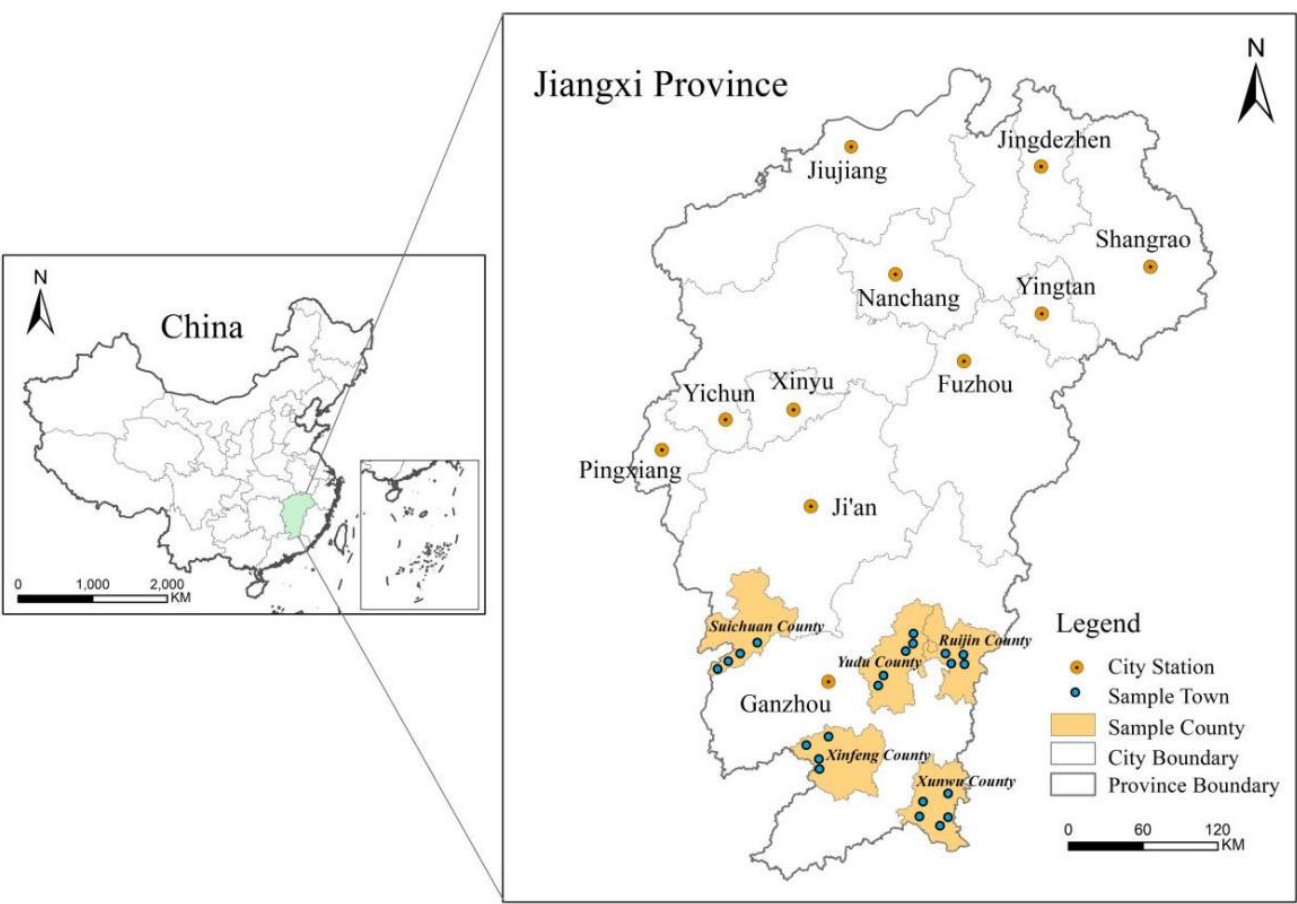

**Figure 2.** Location of Jiangxi Province in China and the study area.

### 4.2.2. Explanatory Variables

The explanatory variable of this paper is the security awareness of land property rights, and the question "Who do you think the land you contracted belongs to?" is used to measure farmers' awareness of land property rights security. We define "ownership by the state" and "ownership by rural collectives" as weak property rights cognition and "ownership by ourselves" as strong property rights cognition. The reason for this division is that when a farmer owns the land and the rights are protected by law, a third party has no right to infringe on the land rights, which can be regarded as "safe"; otherwise, land rights are likely to be infringed upon and can be regarded as "unsafe".

### 4.2.3. Control Variables

Based on the literature, this paper mainly introduces relevant control variables in four dimensions: household characteristics, family characteristics, business characteristics and regional characteristics. The characteristics of the head of household include four variables: gender, age, education level and political identity. Family characteristics include two variables: family asset scale and the number of nonfarm and part-time workers. Management characteristics include three variables: the degree of land fragmentation, the area of cultivated land, and the quality of cultivated land. Regional characteristics include one variable of village location. The definitions and descriptions of the above variables are shown in Table 1.

**Table 1.** Variable definition and statistical description.

| Variable | Variable Definition | Mean | S.D |
|---|---|---|---|
| Applying farmyard manure | Whether to apply farm manure: 1 = Yes; 0 = No | 0.11 | 0.31 |
| Cognition of land property rights security | Strong property rights awareness = 1; Weak property rights awareness = 0 | 0.27 | 0.44 |
| Gender | Gender of head of household: 1 = male, 0 = female | 0.96 | 0.19 |
| Age | Actual age of the head of household, years | 55.90 | 10.02 |
| Education level | 1 = illiterate; 2 = primary school; 3 = junior high school; 4 = high school; 5 = Bachelor's degree or above | 2.64 | 0.85 |
| Political identity | Party member or not: 1 = Yes; 0 = No | 0.16 | 0.36 |
| Household asset scale | Total value of household assets, 10,000 yuan | 25.89 | 28.27 |
| Number of nonfarm workers | Number of migrant workers who have worked for more than 6 months, persons | 1.47 | 1.46 |
| Land fragmentation | Do you think the land you cultivate is too scattered: 1 = Yes; 0 = No | 0.73 | 0.44 |
| Subjective evaluation of cultivated land quality | High-quality cultivated land area, hm$^2$ | 0.12 | 0.53 |
| | Medium-quality cultivated land area, hm$^2$ | 0.41 | 1.97 |
| | Area of low-quality cultivated land, hm$^2$ | 0.03 | 0.12 |
| Village location | Actual distance from village to township government, km | 5.52 | 4.43 |

## 5. Results

### 5.1. Statistical Results

Table 1 shows that the average value of whether to apply farm manure is 0.11, indicating that most farmers do not apply farm manure in the study area. In combination with the subjective evaluation of farmers on the quality of cultivated land, the area of low-quality cultivated land in the study area is also relatively small (the average value of low-quality cultivated land is only 0.03 hm$^2$), and most of the cultivated land quality is at the medium level or above. This indicates that the cultivated land in the study area is not barren and has a certain fertility. Although the terrain of the study area is complex, mainly composed of mountains and hills, the coverage rate of mountain vegetation is high, and the nutritional composition is relatively sufficient, which is different from the barren land in the bare mountains in Northwest China.

The average value of farmers' awareness of land property rights security is 0.27, which is low compared with the number of 669 samples, indicating that farmers' strong awareness of land property rights security in the sample area is still low even after a new round of the land ownership confirmation process. However, its standard deviation is 0.44, which shows that farmers in different counties and cities have large differences in their awareness of land property rights security. Among the household head characteristic variables, the average age is close to 56 years old, and the standard deviation is 10.02. This indicates that the household heads engaged in agricultural production tend to be older. The average number of members of farmer households engaged in part-time work for more than six months is 1.47, which indicates that the phenomenon of migrant workers is common in the study area.

To show the relationship more intuitively between farmers' awareness of land property rights security and investment in farmland protection, a correlation diagram is shown in Figure 3. Among the farmers with strong property rights awareness, 17.88% applied farmyard manure; the proportion of farmers with weak property rights awareness who did so was only 8.00%. This descriptive result roughly shows that there is a positive correlation between the awareness of land property rights security and the behavior of cultivated land quality protection.

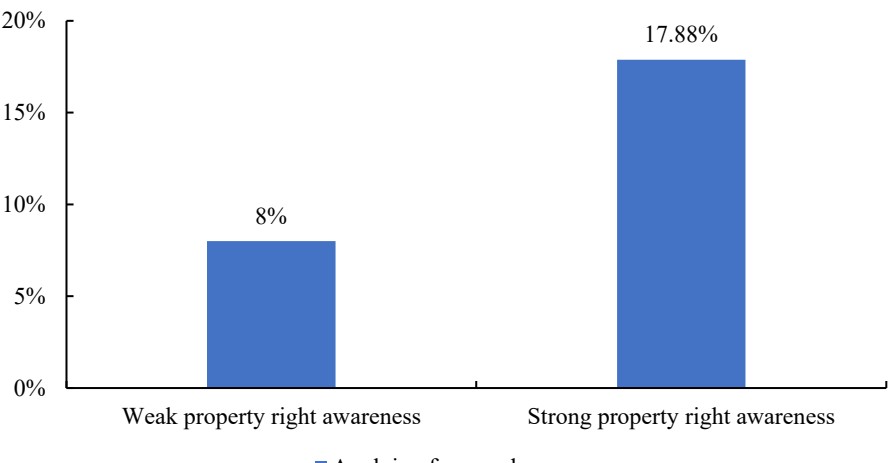

**Figure 3.** Relationship between land property rights security perception and cultivated land quality protection.

## 5.2. Empirical Results

The results of the benchmark regression are shown in Table 2. The results show that farmers' awareness of land property rights security has a significant positive impact on their use of farmyard manure. Compared to farmers with a weak awareness of property rights, farmers with a strong awareness of property rights are more inclined to use farmyard manure. The marginal effect shows that the probability of farmers with strong property rights awareness applying farmyard manure is 9.1% higher than that of farmers with weak property rights awareness. This is consistent with the theoretical expectation of this paper, which verifies the research hypothesis.

**Table 2.** Cognition of land property rights security and farmers' cultivated land quality protection behavior.

| Variable | Whether Farm Manure Is Applied | | |
| --- | --- | --- | --- |
| | **Coefficient** | **Robust Standard Error** | **Marginal Effect** |
| Cognition of land property rights security | 0.519 *** | 0.143 | 0.091 *** |
| Gender | 0.661 | 0.461 | 0.116 |
| Age | 0.011 * | 0.007 | 0.002 * |
| Education level | −0.149 * | 0.090 | −0.026 * |
| Political identity | 0.503 *** | 0.179 | 0.089 *** |
| Household asset scale | −0.001 | 0.002 | −0.001 |
| Number of nonagricultural part-time workers | −0.001 | 0.043 | −0.001 |
| Land fragmentation | 0.035 | 0.152 | 0.006 |
| Subjective evaluation of cultivated land quality (with "high quality" as reference group) | | | |
| Medium quality | −0.005 | 0.004 | −0.001 |
| Low quality | −0.013 | 0.040 | −0.002 |
| Village location | YES | YES | YES |

Note: (1) *** and * indicate significance at the 1% and 10% levels, respectively. (2) Robust standard errors are given in parentheses.

For the control variables, the variables of household head characteristics should be considered. The age, education level and political status of the household head have a significant impact on farmers' application of farmyard manure. Compared with younger farmers, older farmers are more inclined to apply farmyard manure, and the application probability increases by 0.2%. A possible reason is that older farmers have rich planting

experience and understand the significance of farmyard manure to the protection of cultivated land quality. Moreover, older farmers have lower opportunity costs for farming, so they have more time to apply farmyard manure. The education level of the head of household has a significant negative impact on farmers' use of farm manure. With every increase in the education level of the household head (such as changing from junior high school to senior high school), the probability of using farm manure decreases by 2.6%. This may be because a household head with a lower education level has fewer nonagricultural employment opportunities, finds it difficult to understand new ideas, and has a deeper understanding of the protection of soil fertility by farm manure because of long engagement in agricultural production in rural areas. Therefore, these farmers prefer to use farmyard manure. The political status of the head of household also has a significant positive impact on farmers' use of farmyard manure. Household heads with CPC membership are more inclined to use farmyard manure, with a probability of 8.9% compared with household heads who are nonparty members. This is related to party members taking the lead in using farmyard manure and implementing the government's call to protect the quality of cultivated land.

### 5.3. Endogeneity Discussion

As discussed in Section 3.2, the benchmark regression model may have endogeneity problems. To exclude the endogeneity of the model, we introduce the average value of other farmers' perception of land property rights security in the village as the instrumental variable of the core explanatory variable to test the robustness of the model. We find that the Wald test does not pass the significance level test, indicating that farmers' perception of land property rights security is not an endogenous variable. That is, endogeneity is not an issue in the benchmark model.

### 5.4. Robustness Test

To further test the robustness of the regression results of the benchmark model in Table 2, this paper uses the replacement model method and the adjusted variable method. First, we use the replacement model method to replace the probit model with the logistic model for regression. Second, we adopt the variable adjustment method to adjust the explanatory variable "whether farmers use farmyard manure" in two ways. First, we adjust it to "the amount of farmyard manure applied per unit land area" and use the Tobit model for regression. Second, we adjust the "application amount of farmyard manure per unit land area" into classified variables by segments: no use of farmyard manure takes the value 1; an application amount of farmyard manure between 0~15 t/hm$^2$ takes the value 2; an application amount of farmyard manure between 15~30 t/hm$^2$ takes the value 3; an application amount of farmyard manure between 30~45 t/hm$^2$ takes the value 4; and an application amount of farmyard manure above 45 t/hm$^2$ takes the value 5. The second kind of adjusted dependent variable is an ordered classification variable, so the ordered probit (oprobit) model and ordered logit (ologit) model are used for regression. The regression results are shown in Table 3.

The regression results in Table 3 show that, compared with the benchmark regression results in Table 2, the significance and symbols of land property rights security cognition are consistent with those in Table 2, regardless of replacement models or adjustment variables. This fully shows that the regression results in this paper are robust and reliable.

**Table 3.** Robustness test results of the impact of land property security awareness on the amount of farmyard manure applied per unit land area.

| Variable | Replaced Model | Variable Adjustment | | |
|---|---|---|---|---|
| | Logit Model | Oprobit Model | Ologit Model | Tobit Model |
| Cognition of land property rights security | 0.999 *** (0.272) | 0.0495 *** (0.139) | 0.971 *** (0.267) | 89.913 *** (23.409) |

**Table 3.** *Cont.*

| Variable | Replaced Model | | Variable Adjustment | |
|---|---|---|---|---|
| | **Logit Model** | **Oprobit Model** | **Ologit Model** | **Tobit Model** |
| Control variable | YES | YES | YES | YES |
| Village location | YES | YES | YES | YES |
| Prob > chi2 | 0.003 *** | 0.003 *** | 0.002 *** | 0.002 *** |
| Pseudo R2 | 0.066 | 0.044 | 0.045 | 0.024 |
| Sample size | 669 | 669 | 669 | 669 |

Note: (1) *** indicates significance at the 1% level. (2) Robust standard errors are given in parentheses.

## 6. Heterogeneity Analysis

As mentioned above, the differences in individual characteristics, family characteristics and preferences of farmers may lead to different levels of awareness of land property rights security, which will affect the application of farmyard manure. Based on this, this paper conducts a comparative analysis of different generations of farmers, farmers with different part-time jobs, farmers with different land scales and other differences to further investigate the impact of these characteristics and farmers' awareness of land property rights security on the use of farmyard manure.

### 6.1. Comparative Analysis of Farmers in Different Generations

Due to the differences in knowledge, experience, and opportunity costs among different generations of farmers, they pay attention to agricultural production to different degrees. In this study, based on the field survey, heads of household who are older than 55 years old are considered to be the older generation of farmers, and the heads of household who are younger than 55 years old are considered the younger generation of farmers. Table 4 reports the impact of farmers' awareness of land property rights security in different generations on the use of farmyard manure. The results show that for both the younger generation and the older generation of farmers, the improvement in their awareness of land property rights security can significantly increase their probability of using farmyard manure. Regarding the marginal effect, the perception of land property rights security has a greater impact on the use of farmyard manure by the older generation of farmers than by the younger generation of farmers.

**Table 4.** Comparative analysis of farmers in different generations.

| Variable | Whether Farmers Use Farmyard Manure | | | |
|---|---|---|---|---|
| | **Young Generation** | | **Older Generation** | |
| | **Coefficient** | **Marginal Effect** | **Coefficient** | **Marginal Effect** |
| Cognition of land property rights security | 0.579 *** (0.219) | 0.082 | 0.483 ** (0.214) | 0.098 |
| Control variable | YES | YES | YES | YES |
| Prob > chi2 | 0.076 * | - | 0.017 ** | - |
| Pseudo R2 | 0.068 | - | 0.080 | - |
| Sample size | 332 | | 333 | |

Note: (1) ***, ** and * indicate significance at the 1%, 5%, and 10% levels, respectively. (2) Robust standard errors are given in parentheses.

### 6.2. Comparative Analysis of Farmers with Different Part-Time Jobs

The opportunity cost of agricultural production varies with the degree of concurrent employment of farmers, resulting in different attitudes toward the application of farm manure. In this paper, households in which more than 50% of the family members have had outside work for more than six months are considered oriented toward nonagricultural employment. Other households are considered oriented toward both nonagricultural and agricultural employment. The purpose of this distinction is to reflect the different degrees

of part-time employment among farmers. Table 5 reports the comparative analysis of the two types of farmers' awareness of land property rights security regarding the use of farm manure. The results show that the improvement in awareness of land property rights security can encourage both types of farmers to use farm manure. The marginal effect reflects that for farmers who consider nonagricultural employment and agricultural production, the improvement in land property rights security awareness has a greater impact on their use of farm manure.

**Table 5.** Comparative analysis of farmers with different degrees of part-time employment.

| Variable | Whether Farmers Use Farmyard Manure | | | |
| | Mainly Nonagricultural Employment | | Both Nonagricultural and Agricultural Employment | |
| | Coefficient | Marginal Effect | Coefficient | Marginal Effect |
|---|---|---|---|---|
| Cognition of land property rights security | 0.459 ** (0.156) | 0.081 | 0.950 ** (0.063) | 0.141 |
| Control variable | YES | YES | YES | YES |
| Prob > chi2 | 0.031 ** | - | 0.037 ** | - |
| Pseudo R2 | 0.049 | - | 0.247 | - |
| Sample size | 93 | | 572 | |

Note: (1) ** indicates significance at the 5% level. (2) Robust standard errors are given in parentheses.

### 6.3. Comparative Analysis of Farmers with Different Land Scales

This paper further analyzes the impact of farmers' awareness of land property rights security on the application of farmyard manure. According to whether the land management area is greater than the average of the sample, the overall sample is divided into two types of farmers by land scale: large scale and small scale. The regression results are shown in Table 6. Awareness of land property rights security can significantly encourage small-scale farmers to use farmyard manure, while the incentive effect on large-scale farmers is not significant.

**Table 6.** Comparative analysis of farmers with different land scales.

| Variable | Whether Farmers Use Farmyard Manure | | | |
| | Large Scale | | Small Scale | |
| | Coefficient | Marginal Effect | Coefficient | Marginal Effect |
|---|---|---|---|---|
| Cognition of land property rights security | 0.526 (0.472) | 0.064 | 0.557 *** (0.153) | 0.102 |
| Control variable | YES | YES | YES | YES |
| Prob > chi2 | 0.155 | - | 0.002 *** | - |
| Pseudo R2 | 0.109 | - | 0.065 | - |
| Sample size | 113 | | 556 | |

Note: (1) *** indicates significance at the 1% level. (2) Robust standard errors are given in parentheses.

## 7. Conclusions and Discussion

The security or stability of property rights plays an important role in stimulating the investment of economic entities. This paper focuses on the perspective of farmers' awareness of land property rights security and analyzes its impact on farmers' investment in farmland quality protection (using farm manure). The study found that the improvement in farmers' awareness of land property rights security can significantly improve the probability of farmers applying farmyard manure and thus protecting arable land. The robustness of the research results is verified through replacement models, adjustment variables and other approaches, and it is found that the empirical analysis results are robust. Moreover,

the comparative analysis of the farmland quality protection behavior of farmers with different endowment characteristics shows that there are differences in the impact of land property security cognition on the application of farmyard manure by farmers of different generations, with different degrees of part-time employment and with different land sizes.

A literature review shows that there are numerous academic studies on the impact of property stability or security on investment. In fact, the security or stability of property rights is a comprehensive concept that is generally subdivided into legal security, factual security, and perceptual security. From the perspective of scientific falsifiability, some scholars believe that the security or stability of property rights must be considered at the legal and factual levels because it is observable and falsifiable [40]. However, the decision-making process of economic subjects is very complex. The decision-making process is affected not only by objective facts but also by individuals' own cognitive level, ideas, and evaluation of people around them and other factors [39]. The understanding of objective facts is ultimately reflected in their own cognitive level. As mentioned above, the Chinese government basically completed a new round of rights confirmation, registration, and certification activities in 2018 (the completion rate reached 97%). It is generally believed that farmers who own land contract certificates will think that the property rights of land belong to them. However, the survey found that after the new round of confirmation, registration, and certification, only 39% of the farmers believed that the contracted land belonged to them [41], which is enough to show that the objective fact of owning a land contract certificate does not make farmers' cognitive level consistent with the theoretical expectation.

In fact, farmers' perception of land property rights security is related to the specific scenarios or experiences of property rights implementation [34,35]. Since the second round of contracting, the Chinese government has defined the land contract period as 30 years and stated that it should adhere to the principle of "great stability, small adjustment". However, the survey data of 17 provinces show that 63.7% of villages adjusted their land during the second round of contracting, and 34.6% of villages adjusted their land after the second round of contracting. Land expropriation and land disputes occasionally occur. These experiences have reduced the perception of land property rights security, thus affecting farmers' land use behavior. Even if farmers hold a land contract certificate and actually own farmland, if they experience land adjustment, expropriation or disputes, their perception of the security of the land contract certificate will be reduced.

## 8. Recommendations

According to the research conclusions and discussions in this paper, the following two recommendations can be made. First, the formulation of public policies should be not only systematic, correct, and in line with the actual situation but also effectively implemented. Because China has an extensive territory, it is difficult to formulate a policy system that can solve all problems once and for all, especially in the context of rural areas with complex terrain. Therefore, we should coordinate the contradiction between village-level autonomy and land laws and continue to improve relevant legal systems, such as prohibiting land adjustment, reducing land acquisition, and effectively resolving land disputes. In combination with the registration and certification of land ownership, we should enhance farmers' awareness of ownership and the functions of agricultural land property rights and ensure that the implementation of the legal system can help farmers "see and touch", not leaving the property rights system only on paper.

Second, we should increase the knowledge of the ownership and functions of agricultural land property rights and publicize the functions of the land ownership certificate in a language and manner that farmers can understand and accept. We should let farmers truly understand the key points of the policy, such as the fact that the land contractual management right will remain unchanged for a long time and that the newly issued land contractual management certificate can effectively protect farmers' land rights. In combination with the change in publicity means, new media, networks, radio and television and other channels should be used, and nongovernmental organizations should be closely

relied on to play a role. Relevant policies and laws should be publicized, and the single publicity channel relying solely on village leaders should be changed. These steps can effectively improve farmers' cognitive restrictions regarding property rights, improve their cognitive ability and level, correct cognitive bias, and then stimulate their awareness and behavior of farmland quality protection.

**Author Contributions:** L.Z. had the original idea and collected data, H.L. and J.Z. carried out the analyses for the study. All authors have read and agreed to the published version of the manuscript.

**Funding:** This study was financed by the National Natural Science Foundation of China (No. 71803071; 72164014); Jiangxi Provincial Natural Science Foundation (20224BAB205048).

**Data Availability Statement:** The data that support the findings of this study are available from the corresponding author upon reasonable request.

**Conflicts of Interest:** The authors have no conflict of interest to declare.

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
