# Peer review of "Impact of Land Property Rights Security Cognition on Farmland Quality Protection: Evidence from Chinese Farmers"

_land, doi:10.3390/land12010188_

Round 1
Reviewer 1 Report
The article discusses an interesting topic. The driving factors of farmland protection were investigated from the perspective of property security. The author writes fluently and can appeal to a wide range of readers. I suggest publishing it after minor revisions.
(1) In the introduction. I suggest that the author add some data about farmland pollution to highlight the importance of farmland protection.
(2) Literature review. I suggest that the author combine the review and hypothesis as the theoretical analysis of the paper.
(3) Method part. The author adopted instrumental variable method for regression, and I suggest that the authors add the test of instrumental variables.
Author Response
Dear Reviewer,
Thank you very much for giving us an opportunity to revise our manuscript. These comments are all valuable and very helpful for improving our paper, as well as the important guiding significance to our researches. We studied the comments carefully and tried our best to revise the manuscript. Point-by-point responses to the reviewers’ comments are provided below, and changes made on the revised manuscript have been marked by using the track changes mode in MS Word. We hope that you are satisfied with the revised version.
Comment 1: In the introduction. I suggest that the author add some data about farmland pollution to highlight the importance of farmland protection.
Response: Thank you. We have revised the introduction section. The following four papers you suggested have been added in revised manuscript.
- According to Guangming. com, for a long time, due to intensive utilization, the basic fertility of cultivated land has declined. In 2018, the amount of chemical fertilizer was 6.4 times that in 1978, while the grain yield was only 2.2 times. Therefore, the protection of cultivated land quality is imminent.
Example (Line 39-44 in section 1). ...attention from government departments and academia. According to Guangming. com, for a long time, due to intensive utilization, the basic fertility of cultivated land has declined. In 2018, the amount of chemical fertilizer was 6.4 times that in 1978, while the grain yield was only 2.2 times. Therefore, the protection of cultivated land quality is imminent.
Comment 2: Literature review. I suggest that the author combine the review and hypothesis as the theoretical analysis of the paper.
Response: Many thanks for your valuable advice. We have integrated the literature review and hypothesis in the second part, and changed the second part into theoretical analysis and research hypothesis.
Comment 3: Method part. The author adopted instrumental variable method for regression, and I suggest that the authors add the test of instrumental variables.
Response: Many thanks for your valuable advices. We have done the validity test of instrumental variables, See Section 5.3 for the test results.
We appreciate for the Reviewer’s warm work earnestly, and hope that the correction will meet with approval.
Once again, thank you very much for your comments and suggestions.
Yours
Sincerely
Hua Lu

Reviewer 2 Report
some minor grammatical errors and computer printing errors are found. These may be modified after reading the text carefully
Author Response
Dear Reviewer,
Thank you very much for giving us an opportunity to revise our manuscript. These comments are all valuable and very helpful for improving our paper, as well as the important guiding significance to our researches. We studied the comments carefully and tried our best to revise the manuscript. Point-by-point responses to the reviewers’ comments are provided below, and changes made on the revised manuscript have been marked by using the track changes mode in MS Word. We hope that you are satisfied with the revised version.
Comment: Some minor grammatical errors and computer printing errors are found. These may be modified after reading the text carefully.
Response: Thank you. We submitted the manuscript to a professional organization and asked them to help correct the grammar errors. The revision has been completed.
We appreciate for the Reviewer’s warm work earnestly, and hope that the correction will meet with approval.
Once again, thank you very much for your comments and suggestions.
Yours
Sincerely
Hua Lu

Reviewer 3 Report
The literature review is missing complitely the point of commoning and the procesesses of recommonification in western coutries and Latin America, in particular with their implication regarding agroecology. I suggest to take this aspect into consideration and give evidence of the fact that property rights and investements are not automatically connected to soil preservation. At the countrary the very strong property right on EU and USA have lead to exploitation of soil and drastic reduction of fertility and biodiversity. I think that the bad experience in other countries should be taken in serious consideration while suggesting reforms of policy in China.
Author Response
Dear Reviewer,
Thank you very much for giving us an opportunity to revise our manuscript. These comments are all valuable and very helpful for improving our paper, as well as the important guiding significance to our researches. We studied the comments carefully and tried our best to revise the manuscript. Point-by-point responses to the reviewers’ comments are provided below, and changes made on the revised manuscript have been marked by using the track changes mode in MS Word. We hope that you are satisfied with the revised version.
Comment: The literature review is missing complitely the point of commoning and the procesesses of recommonification in western coutries and Latin America, in particular with their implication regarding agroecology. I suggest to take this aspect into consideration and give evidence of the fact that property rights and investements are not automatically connected to soil preservation. At the countrary the very strong property right on EU and USA have lead to exploitation of soil and drastic reduction of fertility and biodiversity. I think that the bad experience in other countries should be taken in serious consideration while suggesting reforms of policy in China.
Response: Thank you. By reading the literature of underdeveloped countries and western countries, we have added some literature on property rights and investment.
Example (Line149-156 in section 2). ...prevent land degradation [28]. Some studies from other fields also found that there is no causal relationship between property rights and investment. For example, The Energy Charter Treaty (ECT) aimed at strengthening property rights did not improve the investment environment [29], the implementation of the rural land regularization project in Benin did not improve the investment in soil fertility [30], and the security reform of forest land property rights in Nicaragua unexpectedly increased deforestation [31]. The academic community......
We appreciate for the Reviewer’s warm work earnestly, and hope that the correction will meet with approval.
Once again, thank you very much for your comments and suggestions.
Yours
Sincerely
Hua Lu
